# Cervical Artery Dissection and Patent Foramen Ovale in Juvenile Stroke: Causality or Casuality? A Familiar Case Report

**DOI:** 10.3390/medsci11020034

**Published:** 2023-05-14

**Authors:** Francesca Antonia Arcadi, Rosa Morabito, Silvia Marino, Caterina Formica, Rocco Salvatore Calabrò

**Affiliations:** IRCCS Centro Neurolesi “Bonino Pulejo”, S.S. 113, Contrada Casazza, 98124 Messina, Italy; francesca.arcadi@irccsme.it (F.A.A.); rosa.morabito@irccsme.it (R.M.); silvia.marino@irccsme.it (S.M.); katia.formica@irccsme.it (C.F.)

**Keywords:** cervical artery dissection, patent foramen ovale, juvenile stroke, cryptogenic

## Abstract

Cervical artery dissection (CAD) and Patent Foramen Ovale (PFO) are important causes of stroke in young patients. Although PFO is considered an independent risk factor for cerebral infarction in young adults with cryptogenic stroke, other concomitant causes may be necessary to cause brain injury. PFO could be a predisposing factor of stroke through several mechanisms including paradoxical embolism from a venous source, thrombus formation in atrial septum, or atrial arrhythmias causing cerebral thromboembolism. The pathophysiology of CAD is poorly understood and includes both constitutional and environmental factors. A causal association is often difficult to establish, as other predisposing factors may also play a role in CAD etiopathogenesis. We present a family with ischemic stroke (a father and his three daughters), in which the two different stroke causes are present. We hypothesized that a paradoxical embolism caused by PFO, associated with arterial wall disease, in the presence of a procoagulant state, could produce arterial dissection and then stroke.

## 1. Introduction

Cervical artery dissection (CAD) and Patent Foramen Ovale (PFO) represent important causes of stroke in young and middle-aged patients [1]. The etiology of spontaneous extra and/or intra cranial arterial dissection is considered multifactorial. Vessel wall weakness is probably present in patients with spontaneous dissection who are also more predisposed to suffer from intracranial aneurysm [2], aortic root dilatation [3], and arterial redundancies [4]. CAD may be determined by several genetic and environmental factors, both with a modest and potentially synergistic effect [5,6]. Monogenic connective tissue diseases, such as Ehlers–Danlos syndrome, osteogenesis imperfecta, and Marfan Syndrome, are frequent in patients with spontaneous extra and/or intracranial arterial dissection [7,8]. However, familiar cases of extra and/or intracranial arterial dissections were reported, also in the absence of connective tissue diseases [9]. The mechanism of stroke in patients with CAD is often unclear. The most probable mechanism seems to be related to the presence of artery embolism, as supported by brain imaging studies that show embolic infarcts in stroke patients affected by arterial dissection [10]. Other possible mechanisms include hypoperfusion that produced watershed infarctions in the presence of severe vessel narrowing, occlusion of dissected vessel or, less frequently, the intimal flap that occlude the ostium of a branch of a dissected vessel [11,12].

Several studies considered PFO as a risk factor for stroke. PFO is a congenital right-to-left interatrial shunt that is highly common in the general population, although it is more frequent in patients investigated for cerebrovascular events of unknown origin. Indeed, it has been found to be present in up to 50% of patients under 55 years old who have had a cryptogenic stroke. PFO is related to several mechanisms including paradoxical embolism from a venous source [13], atrial thrombus formation [14,15], and/or atrial arrhythmias producing thromboembolic cerebral diseases [16]. Although, due to these potential sources of paradoxical embolism, PFO is a widely accepted mechanism of stroke, to confirm such a diagnosis is still complex. In fact, in the absence of direct visualization of thrombus in PFO, the diagnosis is still presumed in the setting of an otherwise cryptogenic stroke. PFO closure to eliminate the shunt is considered the new gold standard therapy, especially in patients with a large atrial septal defect. In fact, a recent systematic review has demonstrated that the septal abnormality closure could be more effective than medical treatment only in a selected group of patients, i.e., in male patients (RR 0.34, 95% CI 0.15–0.75, I^2^ = 36%, *p* = 0.07), in those aged <45 (RR 0.35, 95% CI 0.15–0.79, I^2^ = 0%, *p* = 0.01), and in patients with a large PFO shunt (RR 0.25, 95% CI 0.12–0.54, I^2^ = 0%, *p* = 0.0004).

However, no definitive guidelines are present on the medical treatment option before and after the PFO closure, especially concerning the use of anticoagulants vs. antiplatelets and the duration of the drug intake.

Other findings have focused on the potential role of an underlying prothrombotic state in predisposing young patients with PFO to brain embolism. Indeed, the presence of genetic thrombophilia, as well as acquired coagulation abnormalities, may favor the formation of atrial/septal thrombus with a consequent paradoxical embolism.

In the present study, we report a case series of ischemic stroke in the same family in which different causes that determined cryptogenic stroke at young age coexist. Our focus was to investigate a possible correlation between CDA and PFO in the etiopathogenesis of stroke.

## 2. Case Presentation

This familiar case-series (father and three daughters) with ischemic stroke was investigated in our neurovascular ambulatory. Two daughters, 33 and 35 years old, respectively, came to our observation after an episode of ischemic stroke (respectively 1 and 2 years before our visit). In both patients, ischemic stroke was diagnosed as caused by right carotid dissection. Medical history was negative, but there was a strong familiarity for cerebrovascular diseases. Indeed, the younger daughter (28 years old), 6 months before our visit, reported a Transient Ischemic Attack (TIA) with amaurosis lasting less than 1 h. The father (65 years old) had an ischemic stroke caused by left vertebral artery dissection at the age of 40.

We then decide to retrospectively investigate all patients, through a detailed neurological assessment, Magnetic Resonance imaging (MRI) with MR- angiography, Color Doppler Ultrasound of extracranial arteries (CDU), Contrast Transcranial Color-Coded Sonography (cTCCs) of middle cerebral artery (MCA), transesophageal echocardiography (TEE), and a complete hematological screening. The patients did not undergo transthoracic echocardiography since the positivity for cTCCs indicates the use of TEE.

Patients gave written informed consent to be included in the study and for anonymous data publication.

The neurological assessment showed that all patients had minor focal neurological symptoms, such as mild left hyposthenia (in the two older daughters), and vertigo with mild dysarthria (in the father). General physical examination, including heart rate and blood pressure, was within the normal range (according to the age-related parameters) in all study subjects.

The MRI examination, which was performed by a 3 Tesla MRI scanner (Philips Achieva), shows chronic cerebral ischemia in all patients except the asymptomatic younger daughter (see Figure 1, Figure 2 and Figure 3).

Color Doppler Ultrasound examination of extracranial arteries was performed by using the Ultrasonography system PHILIPs IU22 with linear probe 7.5 Mhz. Notably, a significant increase in flow rate (160 cm/s) in bilateral internal carotid was seen in all patients. However, morphological alterations, such as bilateral kinking of the extracranial internal carotid and vertebral arteries, were observed only in the father (shown in Figure 4).

In addition, we used the same Ultrasonography System equipped with a 2–5 MHz phased array transducer for the cTCCD examination. After the administration of the shaked saline solution (shaked to produce “natural” bubbles as contrast medium) in the antecubital vein and during Valsalva Maneuver (VM), we detected in the left MCA of all patients a micro-embolic signal type “curtain pattern” (shown in Figure 5).

Consequently, to complete the diagnosis, we performed a Trans-Esophageal Echocardiography (TEE) that showed a tunnel like PFO with a size of about 3 mm. During VM, a right to left shunt was elicited in all patients.

After these examinations, the three daughters underwent a surgical closure of cardiac septal alteration. Subsequently, they were treated with dual antiplatelet therapy (i.e., ASA 100 mg, plus clopidogrel, 75 mg) for 6 months and subsequently only clopidogrel as a lifelong secondary prevention. The father was not submitted to PFO closure, and he was on ASA (300 mg/daily) without any reported stroke recurrence. It has also been reported that the 8-year-old son of the older daughter showed a cardiac septal alteration and that he had undergone surgical treatment as well.

Hematological (including platelets, fibrinogen, D-dimer, Protein S and C, LAC, complement proteins, and autoantibodies) and genetic screening showed no relevant abnormalities, but heterozygosis for the MTHFR C677T genotype in the father, and homozygosis mutation in the three daughters. Homocysteine plasma levels were normal in all patients.

## 3. Discussion

To our knowledge, this is the first “familiar” case series in which two different causes (PFO and CAD), which may determine stroke at young age, were present in four individuals of the same family. PFO has been identified as a risk factor for cerebral infarct in young adults with cryptogenic stroke, but this association is one of the most controversial issues in the literature, also because paradoxical embolism is frequently a diagnosis of suspicion. The origin of this cardiac abnormality is currently unknown. Nonetheless, it has been suggested that PFO is a family trait and may contribute to genetic susceptibility to stroke, even though further family studies are needed to confirm these results and to examine the mode of transmission of this septal defect [17,18,19].

CAD was considered a common cause of stroke in young adults but the exact mechanism of stroke in patients with CAD is unclear. Genetic and environmental risk factors are assumed to contribute to the susceptibility to cervical artery dissection. Recent findings [20] showed that rare genetic imbalance affecting different biological functions contributes to the risk of CAD. The diagnosis of these stroke etiologies may be challenging and involves investigations that are consuming and costly. In fact, magnetic resonance imaging (MRI), including diffusion-weighted imaging (DWI), has become the investigation of choice to detect cerebral ischemia in acute stroke, whereas Angiography is essential to confirm the diagnosis of dissection. Over the last several years, a number of studies reported that single cortical and subcortical lesions may be a consequence of stroke associated to the presence of PFO, whilst multiple brain lesions may be dependent on stroke associated to arterial dissection [21]. However, in the light of the present report, it appears complex to find a clear connection between the two different etiologies known to be involved in the pathogenetic mechanisms of stroke. In an attempt to find a correlation between an arterial dissection and the presence of PFO, Ronco and collegues [22], describing a case of a 50-year-old woman with an acute coronary syndrome combined with a renal infarction, hypothesized that a paradoxical embolism throughout the PFO was the cause, producing the renal embolism and the subsequent coronary artery dissection for high blood pressure. Our patients presented anatomical alterations of intra and extracranial vessels that may have been predisposed to CAD, especially in the presence of hypertension. Unfortunately, we are not aware if high blood pressure levels were present in all cases, but they were in the father; indeed, 220/100 mmHg blood pressure was detected at the emergency room. Notably, it has been reported that anatomical alterations were associated with arterial dissection, both in the presence or in the absence of other clinical signs of connective tissue diseases [23,24,25]. We found important alterations in nearly all the patients, but these were not associated with the most common monogenic causes of stroke we investigated, i.e., Fabry’s Disease, fibrodysplasia, and Marfan Syndrome.

Although PFO is considered by some authors as an anatomical variant, and therefore not a potential cause of ischemic stroke [26,27], it is possible that the presence of other factors contributing to a prothrombotic state may have a role and increase the risk of cerebral ischemia. In fact, the presence of a procoagulant state (factor V Leiden, Prothrombin IIa, and MHTFR) is known to be an independent risk factor for stroke at young age [28]. Additionally, a familiar case with cryptogenic stroke, probably due to the interaction between prothrombotic genetic polymorphism and atrial septal defects, was recently reported [29]. The presence of a family history of stroke should lead to the necessity of a screening for genetic diseases in young patients with stroke. Thereby, we concluded that clinical risk factors, differently correlated between each other, albeit under the same pathogenic mechanisms, could contribute to this intriguing familiar case of stroke. Atrial septal abnormalities, such as the PFO, combined with a prothrombotic state, as well as with a particular hemodynamic condition (all patients presented a significant increase in the supra-aortic vessels flow rate), could produce a paradoxical embolism promoting the emergence of cervical artery dissection (CAD) (Figure 6).

However, although the present study suggests a possible correlation between PFO, CAD, and familial predisposition in determining ischemic stroke, more studies in the future are required for verification of these observations.

Figure 6 shows the pathophysiological hypothesis of the familiar ischemic stroke (d), which was probably due to the interaction between the septal (a) and arterial (c) abnormalities in the presence of a procoagulant state (b).

This is why, in our opinion, PFO should be investigated in patients with cryptogenic stroke, and literature guidelines (especially the recent one by the European Society of cardiology) must be followed in order to better manage this septal abnormality [30]. First, confirming that a cryptogenic stroke is secondary to a PFO is not easy and requires extensive investigation. The benefit from PFO closure is dependent on the probability that the index stroke is attributable to the septal defect. Indeed, it has been shown that large superficial cortical infarcts are more likely due to paradoxical embolism [31] PFO-related ischemic strokes are also more prevalent in patients with a risk of paradoxical embolism (RoPE) score > 5. This latter is a 10-point combined score used in patients with a history of prior cerebral infarct or TIA [32]. Patients with a RoPE score > 7 had higher probability of PFO-related cryptogenic stroke than PFO as an incidental finding, and these patients may benefit more from percutaneous closure [30,31,32]. However, a multidisciplinary team discussion, including an interventional cardiologist, imaging cardiologist, and stroke physicians, is fundamental before the intervention. Notably, the best candidates to PFO closure are patients aged less than 60 years with a good functional recovery following stroke (i.e., a modified Rankin scale < 3). Notably, the PFO closure should be associated with antiplatelet therapy since it seems to reduce the incidence of a recurrent stroke when compared with antiplatelets alone. Antiplatelet therapy should be used after PFO device closure, although the duration of the treatment is based on the discretion of the treating clinician. Finally, PFO closure is considered a safe procedure with a low complication rate, even though new onset atrial fibrillation and thromboembolism can occur [30,33].

Another observation which may be worth underlying is that in the present report, the members of this family did not present, at the time of the stroke onset, any of the most common and well-recognized risk factors (hypertension, dyslipidemia, etc.). Hopefully, the present report will contribute to promoting further studies aimed at understanding the mechanisms underlying stroke in young adults, which is certainly associated with high mortality and deep public health implications due to an important loss of productive years of life.

## 4. Conclusions

With this familiar case report, we have further demonstrated the complex interplay in the etiopathogenesis of juvenile stroke. Indeed, in all the family members, PFO has probably contributed to the onset of the cerebrovascular event, although it is not possible to establish with certainty if it was a causal factor or a casual finding. Nonetheless, it is conceivable that the concomitant presence of the arterial wall alterations, the procoagulant state, and the septal abnormality itself was fundamental for the stroke/TIA to occur.

## Figures and Tables

**Figure 1 medsci-11-00034-f001:**
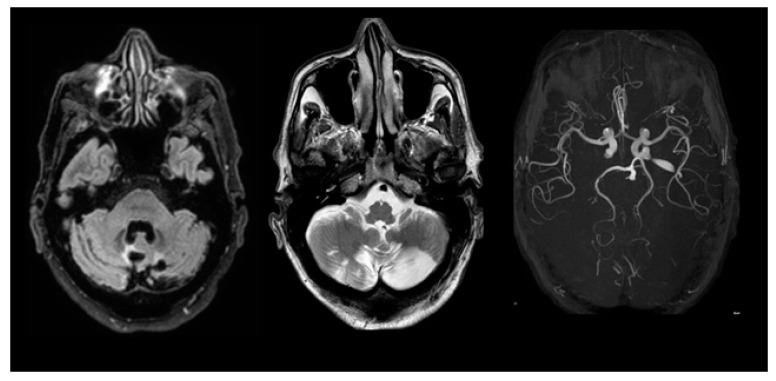
Case 1 (father): MRI examination case 1. (a) axial FLAIR; (b) axial TSE T2. The images show chronic ischemic lesion that involved both the cerebellar hemispheres and the vermis.

**Figure 2 medsci-11-00034-f002:**
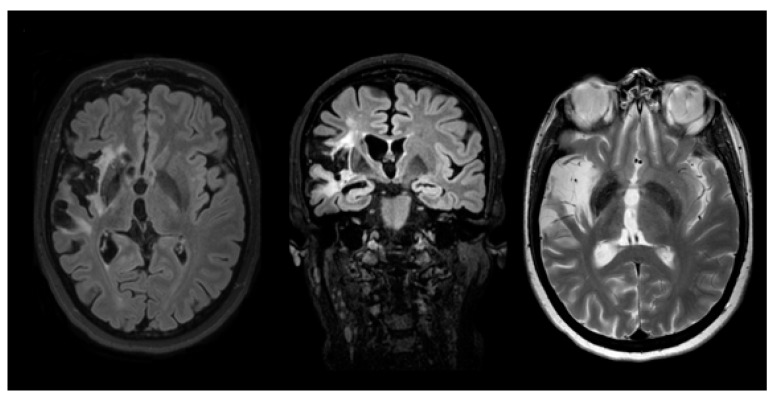
Case 2 (1st daughter): MRI examination case 2. (a) axial FLAIR; (b) coronal FLAIR; (c) axial TSE T2. The images show a chronic infarct involving the frontal, temporal, and insular right lobes.

**Figure 3 medsci-11-00034-f003:**
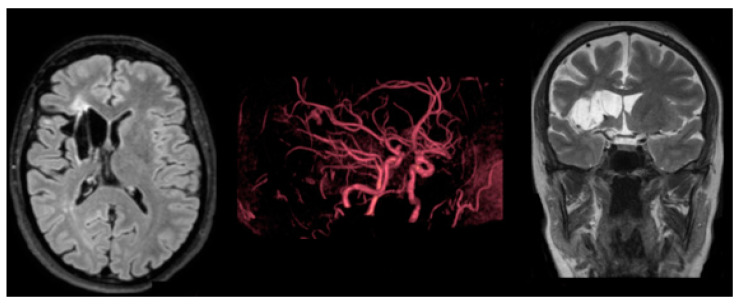
Case 3 (2nd daughter): MRI examination case 3. (a) axial FLAIR; (b) coronal TSE T2; (c) MRA imaging. (a) and (b) show an old ischemic lesion involving the right frontal-insular region, the head of caudate, and the lenticular nuclei. (c) demonstrates a saccular aneurysm of the right internal carotid artery with a diameter of 2.8 mm.

**Figure 4 medsci-11-00034-f004:**
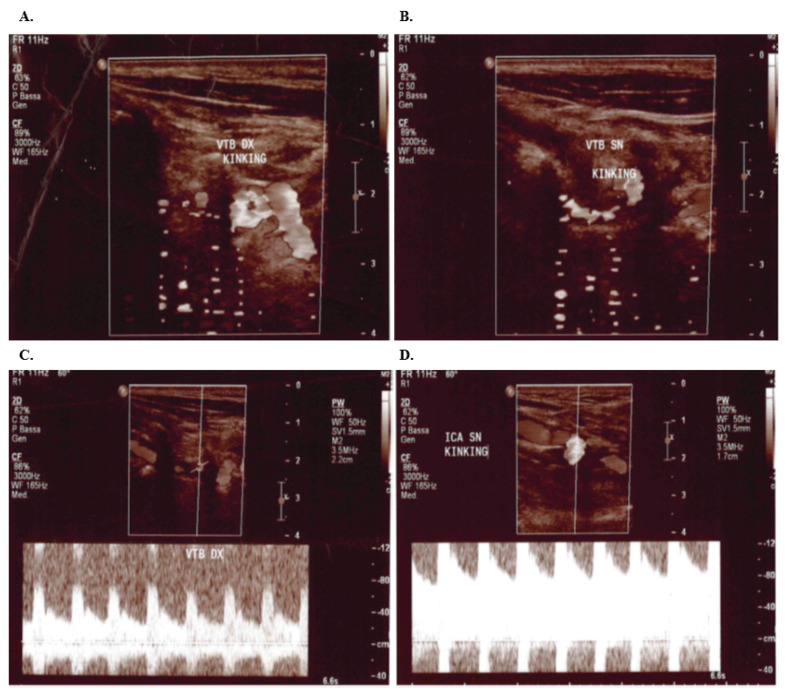
Bilater kinking of the father’s Carotid artery (**A**) kinking of the right vertebral artery; (**B**) kinking of the left vertebral artery; (**C**) increased right vertebral artery flow; (**D**) kinking of the left internal carotid with increased blood flow.

**Figure 5 medsci-11-00034-f005:**
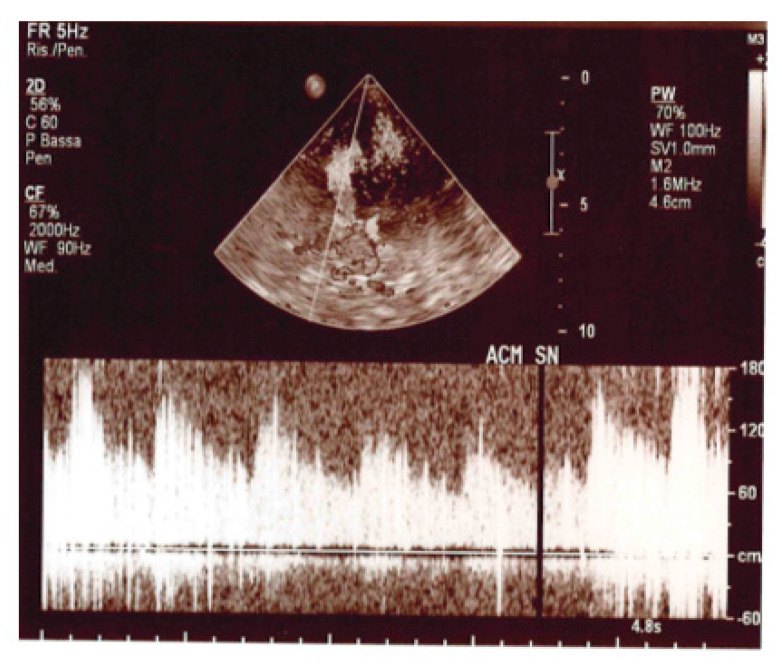
TDC shows the presence of bubble contrast in the left MCA (curtain effect).

**Figure 6 medsci-11-00034-f006:**
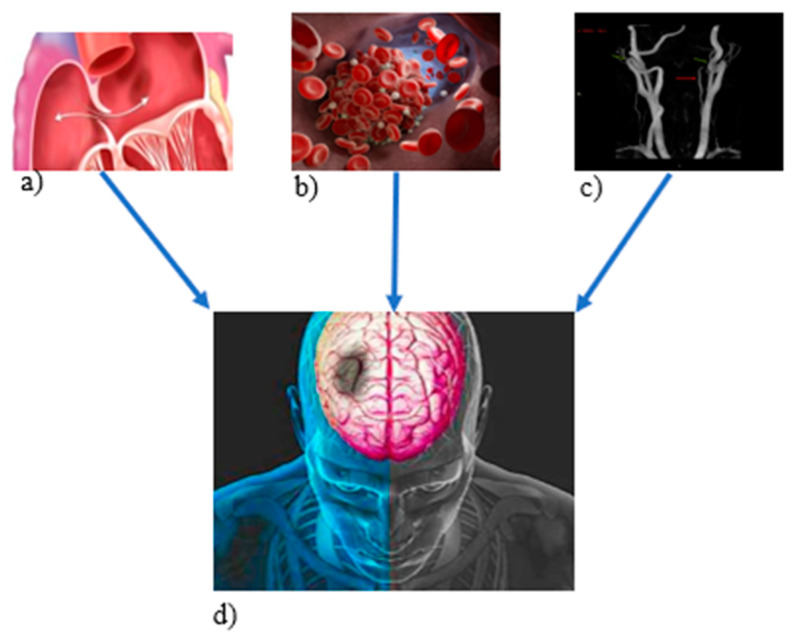
Pathological hypothesis of the multifactorial familiar ischemic stroke (**a**) atrial defect; (**b**) procoagulant state; (**c**) arterial abnormalities; (**d**) ischemic stroke.

## Data Availability

Data are available upon reasonable request by researchers to the corresponding author.

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
