# Peer review of "Cervical Artery Dissection and Patent Foramen Ovale in Juvenile Stroke: Causality or Casuality? A Familiar Case Report"

_medsci, 2023, doi:10.3390/medsci11020034_

Round 1

Reviewer 1 Report

The present case report is timely and of interest for people working in the field. i would reccommend a schematic figure to illustrate the proposed hypothesis.

Author Response

The present case report is timely and of interest for people working in the field. i would reccommend a schematic figure to illustrate the proposed hypothesis.

The figure was added, as suggested

Reviewer 2 Report

Thank you very much for inviting me to review the article entitled "Cervical artery dissection and Patent Foramen Ovale in juvenile stroke: causality or casuality? A familiar case report”.

The authors present “familiar” case series in which two different causes: cervical artery dissection (CAD) and Patent Foramen Ovale (PFO), which may determine stroke at young age, were present in 4 individuals of the same family.

The work is interesting, explores e vary important issue. It contains minor weaknesses.

1.     Could the authors include laboratory tests and include platelets, fibrinogen, coagulation pattern.

2.     I kindly request that the authors complete the case description with a history of previous illnesses, physical examination on admission (blood pressure, heart rhythm), electrocardiography and transthoracic echocardiography.

3.     The authors wrote that three women received dual antifibrotic therapy after PFO closure. Please detail the treatment (substance and dose) and what the further treatment plans were. How long was the dual therapy supposed to last?

4.     Please specify how the man was treated.

5.     In the discussion, please refer to how the literature recommends treatment - this would be useful in clinical practice.

The work is worthy of publication after revision.

Author Response

The work is interesting, explores e vary important issue. It contains minor weaknesses.

  1. Could the authors include laboratory tests and include platelets, fibrinogen, coagulation pattern.

As specified, all tests were within the normal range, but genetic thrombophilia

  1. I kindly request that the authors complete the case description with a history of previous illnesses, physical examination on admission (blood pressure, heart rhythm), electrocardiography and transthoracic echocardiography.

The data have been added, however, TTE was not performed because after the positivity to TDC, patients were directly submitted to TEE, as specified in the text.

  1. The authors wrote that three women received dual antifibrotic therapy after PFO closure. Please detail the treatment (substance and dose) and what the further treatment plans were. How long was the dual therapy supposed to last?

They were treated with dual antiplatelet therapy (i.e., ASA 100 mg, plus clopidogrel, 75 mg) for 6 months and then, only with clopidogrel as lifelong secondary prevention.

  1. Please specify how the man was treated.

The father was not submitted to PFO closure, and was yet on ASA (300 mg/daily) without any reported stroke recurrence.

  1. In the discussion, please refer to how the literature recommends treatment - this would be useful in clinical practice.

Evidence -based Recommendation treatment has been added in the discussion.

Reviewer 3 Report

Dear Authors,

It is impressive the "family" incidence of cervical artery dissection. However, I disagree with the way you decided to approach this topic. Cervical artery dissection is a rare mechanism for stroke pathogenesis, and has to be excluded during work - up of ESUS or cryptogenic strokes (U/S and Doppler, CTA or MRA). Only if these exams are negative, then PFO could be considered as the origin of embolii. Moreover, prevalance of PFO among general population is considered as high (20-30%). So, I am not surprised that among the members of a family, you found out subjects with PFO.

--

Author Response

The reviewer is right with this concern, since we ourselves, yet in the title, have specified that the association could be causal or casual. We have added evidence and treatment recommendation in the discussion, pointing out the cases in which there is more probability of PFO-related stroke.       The novelty of the case description stands in that different concomitant rare causes should be investigated when dealing with young patients with stroke, as better specified.

Round 2

Reviewer 3 Report

Despite the improvements in the manuscript, I disagree with its concept about the correlation between PFO, cervical arterial dissection and familial predispotion.

none